# Enhanced Adsorption of Sulfonamides by Attapulgite-Doped Biochar Prepared with Calcination

**DOI:** 10.3390/molecules27228076

**Published:** 2022-11-21

**Authors:** Jianqiao Hu, Feng Liu, Yongping Shan, Zhenzhen Huang, Jingqing Gao, Wentao Jiao

**Affiliations:** 1College of Water Conservancy and Civil Engineering, Zhengzhou University, Zhengzhou 450066, China; 2Research Center for Eco-Environmental Sciences, Chinese Academy of Sciences, Beijing 100086, China; 3College of Ecology and Environment, Zhengzhou University, Zhengzhou 450066, China

**Keywords:** attapulgite/biochar, sulfadiazine, sulfamethazine, adsorption mechanism

## Abstract

The extensive use of sulfonamides seriously threatens the safety and stability of the ecological environment. Developing green inexpensive and effective adsorbents is critically needed for the elimination of sulfonamides from wastewater. The non-modified biochar exhibited limited adsorption capacity for sulfonamides. In this study, the attapulgite-doped biochar adsorbent (ATP/BC) was produced from attapulgite and rice straw by calcination. Compared with non-modified biochar, the specific surface area of ATP/BC increased by 73.53–131.26%, and the average pore width of ATP/BC decreased 1.77–3.60 nm. The removal rates of sulfadiazine and sulfamethazine by ATP/BC were 98.63% and 98.24%, respectively, at the mass ratio of ATP to rice straw = 1:10, time = 4 h, dosage = 2 g∙L^−1^, pH = 5, initial concentration = 1 mg∙L^−1^, and temperature = 20 °C. A pseudo-second-order kinetic model (R^2^ = 0.99) and the Freundlich isothermal model (R^2^ = 0.99) well described the process of sulfonamide adsorption on ATP/BC. Thermodynamic calculations showed that the adsorption behavior of sulfonamides on the ATP/BC was an endothermic (Δ*H* > 0), random (Δ*S* > 0), spontaneous reaction (Δ*G* < 0) that was dominated by chemisorption (−20 kJ∙mol^−1^ > Δ*G*). The potential adsorption mechanisms include electrostatic interaction, hydrogen bonding, π–π interaction, and Lewis acid–base interactions. This study provides an optional material to treat sulfonamides in wastewater and groundwater.

## 1. Introduction

Since the 1940s, sulfonamides have been widely used to prevent and treat diseases, and livestock industries have also made extensive use of sulfonamides as growth promoters for the growth of farm animals [1]. However, only a small amount of sulfonamides can be adsorbed by the parent, and more than 75% of sulfonamides will be excreted [2], which seriously threatens the safety and stability of the ecological environment. Due to the limited removal efficiency of antibiotics in sewage plants, sulfonamides were frequently detected in soil, organic fertilizer, wastewater, surface water, groundwater, and other environments. The highest concentration of sulfonamides detected in surface water near a pig farm in Jiangsu reached 0.211 mg∙L^−1^ [3]. The average concentration of sulfonamides in surface water in Korea was 20 μg∙L^−1^ [4]. The detection frequency of sulfonamides in groundwater wells in Baden-Württemberg, Germany, was as high as 10%, with a maximum concentration of 410 ng∙L^−1^ [5]. Sulfonamides are difficult to biodegrade [6], and they would form a concentration accumulation effect through biological amplification in water and induce organisms to produce resistance genes, causing more serious harm to the environment. Therefore, the study on the removal of sulfonamides is of great significance for the healthy development of ecological balance.

It is very important to choose a suitable method to effectively solve the pollution of sulfonamides. Many methods have been proposed for the removal of sulfonamides such as ozone oxidation [7], photodegradation [8], electro-Fenton [9], and biodegradation [10]. These techniques have a good effect on the removal of sulfonamides. However, ozone oxidation has a good oxidation effect only for certain kinds of organic substances [7]. Photodegradation technology has the disadvantages of low light utilization rate [8], electro-Fenton is complex in operation [9], and biodegradation requires a long repair period [10], so the use of these technologies is limited by their disadvantages [11,12]. Adsorption is considered as a harmless and simple method for treating antibiotic wastewater [13,14]. Among the commonly used adsorbents, nanomaterials [15] are very effective in removing contaminants from water. However, the high commercial cost [16] limits their use, so it is crucial to find a low-cost and effective adsorbent.

For this reason, biochar, which is cheap and abundant in raw materials, is attracting much attention [17,18]. Biochar (BC) is an easily accessible and environmentally friendly porous material with abundant functional groups and a porous structure, which shows excellent adsorption effects on heavy metals and organic pollutants [19,20]. Many agricultural and industrial waste (sewage sludge, solid waste, food waste, agricultural crop residues, and industrial organic waste) can be made into biochar [21]. Agricultural production accumulates a large amount of agricultural waste such as straws and hulls. The direct burning of these wastes will cause air pollution. Therefore, agricultural waste urgently needs to be treated as a resource, and making adsorbents is a good way of digestion. Previously, pine [22] and waste residue [23] were made into biochar by high-temperature calcination, and the adsorption capacity for sulfonamides ranged from 82.2 to 130 μg∙g^−1^. BC exhibits limited adsorption capacity, so some modification is necessary.

The adsorption capacity and stability of the new material made of clay material and biochar are better than those of the original biochar. The clay–biochar composites from montmorillonite, kaolin, and different biomass sources can increase the adsorption capacity of methylene blue by five times, and the stability of biochar is enhanced due to the increase in stable aromatic carbon [24]. Attapulgite (ATP), also known as palygorskite, is a clay mineral with a water-rich magnesium silicate dominated by a layered chain-like transition structure and belongs to the sepiolite family [25,26]. ATP possesses excellent adsorption performance and carrier performance due to its unique rod crystal morphology and pore structure [27,28].

Theoretically, it is possible to obtain a new adsorbent for sulfonamides by using ATP to modify BC. However, there is a lack of research on the adsorption sulfonamides by BC modified with ATP. It is not clear what the performance of the specific composites is and the involved adsorption sulfonamide mechanism. This is the issue that needs to be further researched. To clarify this point, an attapulgite-doped biochar adsorbent (ATP/BC) was produced from ATP and rice straw by calcination. The main purposes of this research were (1) to study the effects of the raw material ratio on the adsorption capacity of ATP/BC; (2) investigate the adsorption effect of ATP/BC on sulfonamides under the influence of adsorbent dosage, time, solution pH, initial concentration, and temperature; and (3) explain the potential adsorption mechanism by fitting kinetic models, fitting isotherms, and calculating the thermodynamic parameters and some electronic characterization methods.

## 2. Materials and Methods

### 2.1. Reagents

Sulfadiazine (SD, C_10_H_10_N_4_O_2_S, M_W_ = 250.28 g∙mol^−1^) and sulfamethazine (SMZ, C_12_H_14_N_4_O_2_S, M_W_ = 250.28 g∙mol^−1^), purity > 99.9%, were purchased from Sigma Chemical Company, St. Louis, MO, USA. Rice straw was purchased from the Lianfeng Agricultural Company, Lianyungang, China. Attapulgite was collected from Xuyi (Jiangsu, China).

Pretreatment of ATP: 250 mL deionized (DI) water was added into the mixture of 100 g ATP powder and 3 g sodium hexametaphosphate, and then this new mixture was sonicated for 30 min (KQ2200DE ultrasonic cleaner, Kunshan, China) and dried in an oven at 80 °C for 24 h. After drying, ATP was sieved through a 100-mesh screen prior and stored.

### 2.2. Preparation of Adsorbents

The attapulgite-doped biochar adsorbents (ATP/BC) were prepared by adapting the procedure used by Yao et al. [24]. First, a stable ATP suspension was prepared by adding different masses of ATP (0 g, 0.1 g, 1 g, 3 g, 5 g) powder to 100 mL DI water followed by sonication of the mixture for 30 min. Ten grams of rice straw powder was added to the ATP suspension and stirred at room temperature for 2 h to mix it as thoroughly as possible. Subsequently, the ingredients were separated from the mixture and put into an oven at 80 °C for drying. During the preparation of ATP and BC, ATP and rice straw also underwent sonication and drying.

The quartz boat was filled with ATP-treated rice straw powder and then placed in a tube furnace. After 20 min, when an oxygen-free environment was formed inside the tube furnace, the temperature was increased to the target temperature of 700 °C [24] at a rate of 10 °C∙min^−1^ and maintained for 4 h. Rice straw powder without attapulgite modification went through the same calcination process. After the tube furnace cooled to room temperature, all samples were taken out and placed in 200 mL of 0.5 mol∙L^−1^ hydrochloric acid for 30 min to remove the ash. Then, the samples were repeatedly rinsed with DI water and ethanol until they were neutral, and dried in an oven at 80 °C for 24 h. After drying, the samples were passed through a 100-mesh sieve and transferred to a self-sealing bag. The adsorbents prepared in different proportions of raw materials were named BC, ATP/BC-0.01, ATP/BC-0.1, ATP/BC-0.3, and ATP/BC-0.5, respectively.

### 2.3. Characterization of Adsorbent

In this study, the surface morphology of the samples was observed by a field emission scanning electron microscope (Hitachi Su-8020, Tokyo, Japan). The specific surface area and pores of the samples were measured using the Brunauer–Emmett–Teller method (ASAP2020HD88, Atlanta, GA, USA). The crystal structure of the samples was observed by X-ray diffraction (PANalytical X’Pert 3 powder, Heracles Almelo, The Netherlands). A Fourier transform infrared spectrometer (Nicolet IS10, Waltham, MA, USA) was used to analyze the functional groups and chemical bonding information of the samples in the range of 400–4000 cm^−1^.

### 2.4. Sulfonamides Adsorption Experiments

In this study, the intermittent adsorption method was adopted to investigate the influence of dosage (0.2–2 g∙L^−1^), pH (3–11), time (0.17 min–24 h), temperature (20–40 °C), and initial concentration (0.1–2 mg∙L^−1^) of the solution on the final adsorption effect by changing the experimental conditions. The specific operation was to measure 50 mL sulfonamide solution into 68 mL digestion vessels, adding 100 mg adsorbent to maintain the initial concentration of antibiotics of 1 mg∙L^−1^, time of 4 h, pH of 5, and the temperature of 20 ± 1 °C. The samples were withdrawn from a mechanical shaker and immediately filtered through 0.22 μm nylon membrane filters (GE cellulose nylon membrane) to determine the adsorbed sulfonamide concentrations by a UV–Vis spectrophotometer (Agilent Cary 60, Santa Clara, CA, USA). To investigate the effect of pH on the adsorption effect, the pH of the sulfonamide solution was adjusted using 0.1 mol∙L^−1^ HCl or NaOH solutions, and the pH was measured using a pH meter. All of the experimental treatments were performed three times in duplicate and the average values were reported. Additional analyses were conducted whenever two measurements showed a difference larger than 5%. The results of the pre-experiments showed negligible losses from the bottle adsorption, volatilization, and photodegradation. The experimental results of the reaction time were used to fit the adsorption kinetic model, and the experimental results of the sulfonamide initial concentration and reaction temperature were used to fit the isotherm and study the adsorption kinetics. The removal rate η of the antibiotics by the adsorbent and the adsorption capacity $q_{e}$ (mg∙g^−1^) can be calculated from Equations (1) and (2), respectively.
(1)$$\eta =\left(C_{0}-C_{e}\right)/C_{0}\times 100\%$$
(2)$$q_{e}=(C_{0}-C_{e})/M\times V\;$$
where $C_{0}$ (mg∙L^−1^) is the initial concentration of sulfonamides in the solution; $C_{e}$ (mg∙L^−1^) is the equilibrium concentration of sulfonamides in the solution after adsorption; V (mL) is the volume of sulfonamides solution; and M (g) is the mass of adsorbent added.

### 2.5. Adsorption Kinetic and Thermodynamic

In order to investigate the controlling mechanism of the adsorption process of the adsorbent for sulfonamides, a linear fit of the adsorbent adsorbed sulfonamides was performed using the pseudo-first-order kinetic model, pseudo-second-order kinetic model, Elovich kinetic model, and diffusion model [29]:

Pseudo-first-order kinetic model:(3)$$ln\left(q_{e}-q_{t}\right)=lnq_{e}-k_{1}t$$

Pseudo-second-order kinetic model:(4)$$t/q_{t}=t/q_{e}+t/\left(k_{2}q_{e}^{2}\right)$$

Elovich kinetic model:(5)$$q_{t}=\alpha +\beta lnt$$

Diffusion model:(6)$$q_{t}=k_{d}t^{0.5}+C_{1}$$
where $q_{e}$ is the amount of sulfonamide adsorbed onto the adsorbent at equilibrium (mg∙g^−1^); $q_{t}$ is the amount of sulfonamide adsorbed at the instantaneous moment (mg∙g^−1^); $k_{1}$ is the rate constant of the pseudo-first-order kinetic model (min^−1^); $k_{2}$ is the rate constant of the pseudo-second-order kinetic model (mg∙g^−1^·min^−1^); $k_{d}$ is the diffusion rate constant (mg∙g^−1^·min^−1/2^); *α*, *β* are the constants of the Elovich kinetic equation; and $C_{1}$ is the intercept, which is related to the thickness of the boundary layer.

The adsorption isotherm is important for describing how the adsorbate molecules are distributed between the liquid and the solid phases under an equilibrium state. The experimental results of sulfonamide adsorption by the adsorbent at different temperatures were fitted by the Freundlich isotherm and Langmuir isotherm, respectively, and the model equations were as follows [30]:

Freundlich isotherm:(7)$$lnq_{e}=lnk_{f}+nln\;C_{e}$$

Langmuir isotherm:(8)$$C_{e}/q_{e}=C_{e}/q_{max}+1/k_{l}C_{e}$$
where $k_{f}$ is the Freundlich affinity coefficient (mg^(1−*n*)^∙L*^n^*∙g^−1^); *n* is the Freundlich linearity constant, which is related to the adsorption strength; $k_{l}$ is the Langmuir bonding term related to interaction energies (L∙mg^−1^); $C_{e}$ is the concentration of the solution at equilibrium (mg∙L^−1^); and $q_{max}$ is the maximum adsorption capacity (mg∙g^−1^) fitted by the Langmuir isotherm.

The thermodynamic parameters (Δ*G*) (kJ∙mol^−1^), the enthalpy change (Δ*H*) (kJ∙mol^−1^) and entropy change (Δ*S*) (J∙mol∙K^−1^) for the adsorption process were calculated from the following equations [31]:Δ*G* = −*RT*∙*lnK*(9)
(10)lnK=ΔS/R−ΔH/(RT)
where *R* is the gas constant (8.314 J∙K∙mol^−1^); *T* is the absolute temperature (K); and *K* is the equilibrium constants (m∙M^−1^). Δ*H* and Δ*S* can be calculated from the slope and intercept of the linear plot of *lnK* vs. *T*^−1^.

## 3. Results and Discussion

### 3.1. Characterization of Adsorbent

Figure 1 shows the scanning electron microscopy images of ATP, BC, and ATP/BC after magnification of 20,000× and 50,000×. Figure 1a,d shows the morphology of ATP, and many rod-like crystal structures could be observed on the surface of ATP. The SEM images of BC (Figure 1b,e) showed that the surface of BC was smoother and there were abundant ridges. The SEM images of ATP/BC (Figure 1c,f) showed rod-like crystal structures that appeared on the surface of ATP/BC and the overall appearance was umbrella-like after the addition of ATP, indicating the formation of new structures on the surface of BC.

The BET method was measured using the specific surface area and pore width of the adsorbents (BC, ATP, ATP/BC-0.01, ATP/BC-0.1, ATP/BC-0.3, and ATP/BC-0.5). In addition, the BET adsorption–desorption isotherm and pore size distribution data are available in the Appendix A (Appendix A). Table 1 shows that the specific surface area of the ATP/BC composites increased by 73.53–131.26% compared to BC, with the largest specific surface area of 113.75 m^2^∙g^−1^ for ATP/BC-0.1. Furthermore, the addition of ATP decreased the average pore width of the ATP/BC composites by 1.77–3.60 nm, and the average pore width of ATP/BC-0.1 was the lowest, which was 3.01 nm.

Figure 2 shows the XRD spectra of ATP, BC, and ATP/BC. The XRD characteristic peak shape is non-crystalline if it is a bun peak, and crystalline if it is a sharp peak [32,33]. The characteristic peak of BC was a bun peak, which means that the BC itself did not have a crystalline structure. While the characteristic peak of ATP/BC appeared as sharp peaks, it showed that the addition of ATP gave the composite a crystal structure. The characteristic peak at 2θ values of 8.53°, 19.92°, 21.02°, 26.79°, 31.08°, 35.07°, 41.27°, 45.07°, and 50.65° were found in the XRD spectra of ATP. The peaks were close to the JCPDS (Joint Committee on Poder Diffraction Standards) of attapulgite (PDF No. 82-1873). In the XRD spectra of ATP/BC, some peaks disappeared after high-temperature calcination, but there were still characteristic peaks at 2θ values of 8.61°, 19.94°, 21.09°, 26.83°, 35.15°, and 50.30°, which were close to the JCPDS of attapulgite (PDF No. 82-1873). The XRD analysis indicated that the ATP characteristic peak appeared in ATP/BC, and the SEM showed that a new structure appeared in ATP/BC before that. These prove that the ATP was successfully loaded onto the surfaces of the BC.

The FTIR spectra of the BC and ATP/BC before and after the adsorption of sulfonamides are given in Figure 3. The peaks around 3420 cm^−1^ refer to O–H bond stretching vibration on both the BC and ATP/BC surfaces, and this functional group is derived from cellulose, hemicellulose, and lignin in biomass feedstock [34]. ATP/BC shows a vibration peak at 1614 cm^−1^ referring to the C=O bond in the carboxyl group [35]. Both BC and ATP/BC showed strong peaks around 1080 cm^−1^, which can be assigned to C–O stretching vibrations in esters and ethers [23]. The peaks near 800 cm^−1^ should be a C–C bond, indicating the possible presence of olefins and aromatic functional groups inside the adsorbents [36]. The peaks at around 460 cm^−1^ were deemed to be the Si–O stretching vibration [30]. Compared with the FTIR spectra of ATP/BC, the peaks at 3433 cm^−1^ and 1081 cm^−1^ moved to the left and there was no peak at 1614 cm^−1^ in the BC FTIR spectra, which indicated that the structure of BC was changed by the addition of ATP.

### 3.2. Synthesis of ATP/BC

As shown in Figure 4, a small amount of ATP could significantly improve the adsorption performance of the adsorbent, the adsorption capacity of SD, and SMZ by the adsorbents gradually increased when the ratio of ATP to rice straw was lower than 0.1 and showed a decreasing trend when it was higher than 0.1. The adsorption capacity peaked at the ratio of 0.1, and the removal rates reached 73.63% and 68.24%, respectively, compared with pure biochar, and the removal rates increased by 20.31% and 23.26%, respectively.

The trend in the specific surface area and average pore width of the adsorbents with the addition of ATP (Table 1) was similar to that of the adsorption effect, in which the specific surface area of ATP/BC-0.1 was the largest and the average pore width was the smallest, indicating that the adsorption effect was closely related to the specific surface area and average pore width. The increase in specific surface area means that the adsorbent can provide more adsorption sites, which is conducive to the reaction. Furthermore, it is generally believed that [37] macropores (>50 nm) have less effect on adsorption; mesopores (2–50 nm) are channels for contaminants to enter micropores (<2 nm) and their distribution has a greater effect on the adsorption capacity of adsorbents; and micropores are the main sites for contaminant adsorption by the adsorbents. The decrease in the average pore width indicates the increase in the number of mesopores and micropores, which is beneficial to improving the adsorption performance of biochar [38,39]. To better understand the key factors controlling the differences in adsorption affinity among the three adsorbents, the surface-area-normalized adsorption capacity was compared. The order of adsorbent surface-area-normalized adsorption capacity from large to small was BC, ATP/BC-0.01, ATP/BC-0.1, ATP/BC-0.3, and ATP/BC-0.5. The adsorption affinity of BC was better than that of ATP/BC. This result shows that biochar plays a major role in the adsorption process, and ATP improved the adsorption capacity of BC mainly by increasing the specific surface area.

When the ratio of ATP to rice straw was further increased, the specific surface area of the ATP/BC composites started to decrease and the removal rate of sulfonamides also decreased. The reason is that ATP itself is hydrated magnesium silicate crystals, and with the increase in the proportion of ATP, some metal elements in ATP may block the pore channels, leading to the decrease in the specific surface area of the adsorbents. In addition, BC is the main adsorption agent in the composites and ATP plays an improvement and modification role on BC, so ATP itself has a general adsorption effect on sulfonamides, and excessive ATP will result in the decrease in BC content per unit mass, thus affecting the overall adsorption effect of the ATP/BC composites. Because the ATP/BC-0.1 showed the best adsorption effect on sulfonamides, the subsequent experiments were conducted using ATP/BC-0.1. 

### 3.3. Adsorption Performance

Effects of solution pH. Under the acidic condition (Figure 5a), the adsorption effect of ATP/BC-0.1 on SD was better, in which the removal rate of SMZ reached the highest at pH = 5, and the removal rate of SD was nearly the same at pH = 3 and pH = 5. The optimal solution pH for the adsorption of sulfonamides by ATP/BC-0.1 can be determined as 5. The adsorption effect of ATP/BC-0.1 on sulfonamides showed a linear decrease under alkaline conditions. pH can greatly affect the adsorption capacity of ATP/BC-0.1 because sulfonamide is an amphoteric compound [40]. The ionization of the amino and sulfonamide groups on the benzene ring of sulfonamides allows them to exist in a complex mixture of cationic, anionic, and neutral molecules in solutions at different pH values. The pH_pzc_ of ATP/BC-0.1 was found to be 2.7 (Figure 1f). According to the PZC theory, at pH values above the pH_pzc_ value, the adsorbent is negatively charged, and vice versa for pH values below pH_pzc_ [41]. In this experiment, ATP/BC-0.1 was negatively charged. Under acidic conditions, the surface of the sulfonamides combines more hydrogen atoms in the cationic state, which is favorable for the adsorption of sulfonamides by ATP/BC-0.1 through electrostatic gravitational force. The proportion of the anionic form of sulfonamides increases under alkaline conditions, which leads to an electrostatic repulsive effect between sulfonamides and ATP/BC-0.1, resulting in a rapid decrease in its adsorption capacity. Therefore, the pH was adjusted to about 5, which could ensure the best adsorption of sulfonamides by ATP/BC-0.1.

Effects of adsorbent dosage. The removal rates of SD and SMZ by ATP/BC-0.1 improved obviously with the increase in the dosage, and the removal rates of SD and SMZ reached 98.47% and 98.21%, respectively, when the dosage of ATP/BC-0.1 was 2 g/L (Figure 5b). On the other hand, the increase in the dosage also led to the decrease in the adsorption of SD and SMZ by ATP/BC-0.1 from the initial 3.42–3.65 mg∙g^−1^ to 0.48–0.49 mg∙g^−1^. This is because the increasing dosage provides more reaction opportunities for adsorption [42]. The removal rate of sulfonamides by ATP/BC-0.1 gradually increased. In contrast, the concentration gradient between the adsorbents and adsorbates narrowed with more added ATP/BC-0.1, resulting in a reduction in antibiotics absorbed by the unit adsorbent [43]. It should be noted that the decrease in adsorbing capacity with dosage increase originates from excess adsorption centers to the number of antibiotic molecules. However, the adsorbent did not reach the saturation adsorption capacity at this time.

Effects of initial sulfonamide concentration. The removal rate of sulfonamides by ATP/BC-0.1 decreased gradually with the increase in the initial sulfonamide concentration, but the adsorption capacity increased in a linear manner (Figure 5c). This is because a certain mass of adsorbent can provide the same number of adsorption sites [44]. The adsorption sites are not fully utilized at low initial concentrations. As the initial concentration increases, the utilization of adsorption sites increases and the unit adsorption capacity gradually increases.

Effects of adsorption time. With the increase in time, the removal rate of SD and SMZ by ATP/BC-0.1 became higher and higher (Figure 5d). In the initial 0–1 h stage of adsorption, the removal rate and adsorption capacity increased rapidly. The removal rates of SD and SMZ were 71.62% and 69.23% at 4 h, respectively. After 4 h, the whole adsorption process came to a stable stage, and the removal rates and adsorption capacity were stable. The reason for this phenomenon is that the adsorption sites that can be provided on ATP/BC-0.1 were sufficient, and the concentration of sulfonamides in the solution was large at the beginning of the reaction, so the sulfonamides quickly approached the adsorbent, thus the adsorption capacity increased rapidly at the beginning of the experiment. However, as the experiment progressed, the concentration of sulfonamides decreased, leading to a slower diffusion rate, and the number of free adsorption sites decreased on the adsorbent, so the adsorption rate slowed down [45]. After the adsorption equilibrium, the removal rate increased slightly, but not significantly, so it can be considered that the adsorption of sulfonamides reached completion at 4 h.

Effects of reaction temperature. In addition, the effect of the reaction temperature on the adsorption of sulfonamides by ATP/BC-0.1 was also investigated in this study (Figure 5e). It was found that the changes in the removal rate and adsorption capacity of sulfonamides by ATP/BC-0.1 were not significant with the increase in temperature, and the variation was within the error range.

In this study, RDA analysis was conducted to investigate the association between the removal rate, adsorption capacity, and each environmental factor, in order to explore the contribution of each environmental factor to the adsorption effect in the adsorption process, and the experimental results are shown in Figure 6. The results show that the correlation between each environmental factor and the adsorption capacity of ATP/BC-0.1 was in the order of pH (contribution = 64.0%, F = 35.0, *p* = 0.002), dosage (contribution = 24.6%, F = 23.0, *p* = 0.002), initial concentration (contribution = 8.2%, F = 10.3, *p* = 0.006), and time (contribution = 3.2%, F = 4.0, *p* = 0.054). The above factors contributed more to the adsorption effect and temperature (contribution < 0.1%, F < 0.1, *p* = 0.926) had almost no effect on the overall adsorption effect. Therefore, when using ATP/BC-0.1 for the removal of sulfonamides, the focus should be on controlling the pH of the environment and the adsorbent dosage.

Previously, Zhang [23] used coffee grounds to make BC, and the maximum adsorption capacities of BC for sulfadiazine and sulfamethoxazole were 121.5 μg∙g^−1^ and 130.1 μg∙g^−1^ at 25 °C with the initial antibiotic concentration of 500 μg∙L^−1^, respectively. In this study, the adsorption capacities of ATP/BC-0.1 for SD and SMZ were 2.06 mg∙g^−1^ and 1.96 mg∙g^−1^ under the same conditions, respectively. Compared with the biochar from coffee grounds, the adsorption capacity of ATP/BC for sulfonamides increased by about 17 times, which proves that ATP/BC has broad application prospects.

### 3.4. Adsorption Kinetics

The results of the fitting of each adsorption kinetic model for sulfonamide adsorption by ATP/BC-0.1 are shown in Figure 7, and the specific model parameters of each equation are shown in Table 2.

The calculated R^2^ values from non-linear regression (Table 2) indicated that the best fitting order of kinetic model was determined to be pseudo-second-order > pseudo-first-order > Elovich. In addition, the equilibrium adsorption capacity of SD and SMZ predicted from the pseudo-second-order kinetic model equations were 3.55 mg∙g^−1^ and 3.49 mg∙g^−1^, respectively, which were close to the actual ones. In order to further study the law of sulfonamide adsorption by ATP/BC-0.1, the diffusion equation was used to fit the adsorption process (Figure 7d), and the specific model parameters of the equation are shown in Table 2. The process of sulfonamide adsorption by ATP/BC-0.1 mainly includes two stages. Sulfonamides are diffused within the pore space, and as time passes, more adsorption sites are occupied and the solution concentration decreases, so the diffusion resistance will keep increasing, resulting in a smaller k [46]. The curve in the figure did not pass through the origin, implying that, in addition to pore space, diffusion is not the only factor controlling the actual adsorption rate, and there are remaining limiting factors [47].

### 3.5. Adsorption Isotherm

The fit of each adsorption isotherm for the adsorption of sulfonamide by ATP/BC-0.1 is shown in Figure 8, and the specific model parameters of each equation are shown in Table 3.

Although the two models both described the experimental data well, the simulations of the Freundlich model fitted the isotherms better (Table 3). The Freundlich exponent, *n* indicates the affinity of the adsorption. Since *n* > 1, it indicates that ATP/BC-0.1 has a better affinity for sulfonamides and the adsorption easily proceeded [48]. In the Langmuir isotherm, it assumes the presence of a large number of adsorption active centers inside the adsorbent. When all the adsorption active centers are fully occupied, the adsorption is in saturation, and the Langmuir isotherm describes the monolayer adsorption. The Freundlich equation describes the adsorption from low and medium concentrations, when the monolayer is not filled, and the parameter *n* describes the heterogeneity of the adsorption sites [49].

### 3.6. Adsorption Thermodynamics

The relevant thermodynamic parameters during the adsorption of sulfonamides by ATP/BC-0.1 are shown in Table 4. As can be seen from Table 4, all Δ*G* values were negative for sulfonamide adsorption by ATP/BC-0.1, indicating that the adsorption reaction is spontaneous. The higher |Δ*G*| represents the higher driving force of the adsorption process. The table shows that there was a tendency for |Δ*G*| to increase with the temperature rise, but not significantly, so the temperature was not the main factor affecting the adsorption effect. Furthermore, 0 > Δ*G* > −20 kJ∙mol^−1^ represents physisorption while the −20 kJ∙mol^−1^ > Δ*G* represents chemisorption [45]. Therefore, the adsorption mechanism is directed by chemisorption, which involves covalent bonding through sharing or exchanging electrons between the adsorbent and the adsorbate [30]. It is evident that the surface functional groups of the ATP/BC-0.1 surface can promote interaction with the ionic moieties of sulfonamide molecules through valency forces. The adsorption of sulfonamides by ATP/BC-0.1 is elevated as the chemical interactions are further enhanced due to the high surface area and additional functional groups imparted on the pristine BC by ATP [45]. The positive values of ΔS reflect a higher affinity of the ATP/BC-0.1 surface toward the sulfonamide molecules, suggesting the random behavior of the system [50]. All the Δ*H* values were positive, suggesting an endothermic adsorption process [43]. In general, the adsorption behavior of sulfonamides on the ATP/BC is an endothermic, random, spontaneous reaction that is dominated by chemisorption.

### 3.7. Potential Adsorption Mechanisms

According to the results gained from the FTIR, adsorption kinetics, and thermodynamic studies, the adsorption removal mechanisms of sulfonamide by ATP/BC-0.1 mainly involve four removal pathways occurring simultaneously (Figure 9):(1)Electrostatic interactions are an important mechanism to control antibiotic adsorption on carbon materials [45,48]. Under acidic conditions, the surface of sulfonamides binds more hydrogen atoms to present a cationic state and adsorbs with ATP/BC-0.1 by electrostatic interaction. The adsorption effect of ATP/BC-0.1 on sulfonamides is strongly influenced by pH, confirming that electrostatic interactions contribute to the adsorption process.(2)Hydrogen bonding exists in the adsorption process of polar organic pollutants. The ATP/BC-0.1 surface is rich in hydroxyl groups and the benzene ring is the basic structure of sulfonamides. O–H can act as hydrogen bonding donors to form hydrogen bonds with carbon materials, whereas the benzene rings serve as hydrogen bonding acceptors. The O–H vibration at 3418 cm^−1^ moves to lower absorbance areas upon adsorption, which can be attributed to the hydrogen bonding interaction between sulfonamide molecules and these functional groups [51], and Lang reached a similar conclusion when studying the adsorption of antibiotics by phosphorylated alkali lignin [52].(3)π–π interactions are considered to be the driving force for the adsorption of organic chemicals on carbon materials [41]. The amino and sulfonamide groups in sulfonamides act as π-electron acceptors, π orbitals formed by atoms on ATP/BC-0.1 act as π-electron donors, and π–π interactions are generated between ATP/BC-0.1 and sulfonamides for adsorption.(4)There may be Lewis acid–base interactions in the adsorption process [45]. In Figure 3, the peak of ATP/BC-0.1 at 1614 cm^−1^ indicates the presence of a C=O bond with carboxyl groups, and the amino group in sulfonamides can react with carboxyl groups to form Lewis acid–base interactions.

In addition, van der Waals forces also assist the adsorption process. These multiple interactions are beneficial to improve the adsorption capacity of the material.

## 4. Conclusions

An adsorbent consisting of attapulgite and biochar was prepared. ATP/BC inherits the functional groups of biochar. At the same time, ATP/BC has a larger specific surface area with the addition of ATP. Therefore, the adsorption capacity of ATP/BC for sulfonamides is far better than that of BC and ATP alone. Research shows that the adsorption capacity of the ATP/BC for SAs is best at the mass ratio of ATP to rice straw = 1:10, and acid conditions are more conducive to the adsorption of sulfonamides by ATP/BC. The adsorption performance of ATP/BC can be further improved by modulating various environmental factors, and the removal rate of sulfonamides by ATP/BC could reach 98% under the experimental conditions. The thermodynamic results show that the adsorption of sulfonamides by ATP/BC is a heat-absorbing, stochastic, and spontaneous reaction process, and the main adsorption mechanisms include electrostatic interactions, hydrogen bond, π–π interactions, and Lewis acid–base interactions. In general, ATP/BC can be used as an effective adsorbent for the removal of antibiotic contamination from wastewater and groundwater.

## Figures and Tables

**Figure 1 molecules-27-08076-f001:**
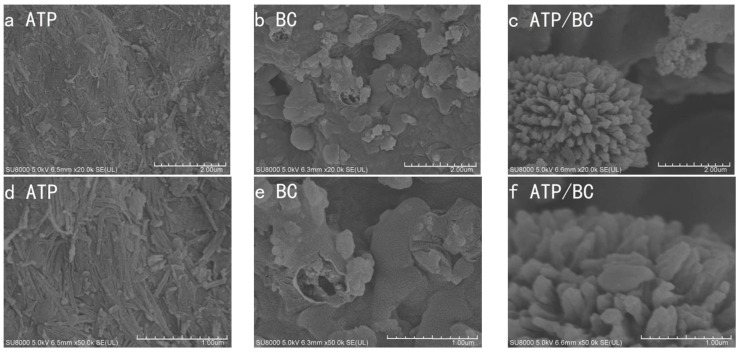
Scanning electron microscope images of ATP (**a**,**d**), BC (**b**,**e**), and ATP/BC (**c**,**f**).

**Figure 2 molecules-27-08076-f002:**
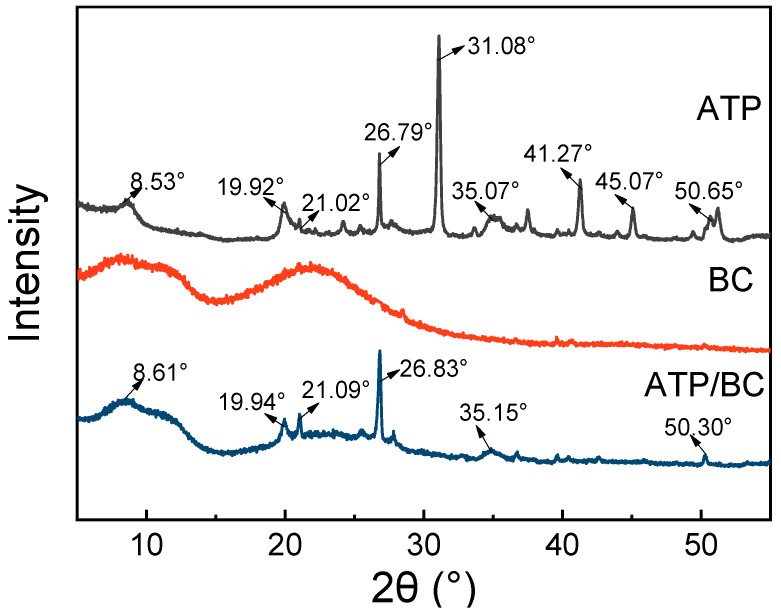
The XRD spectra of the ATP, BC, and ATP/BC.

**Figure 3 molecules-27-08076-f003:**
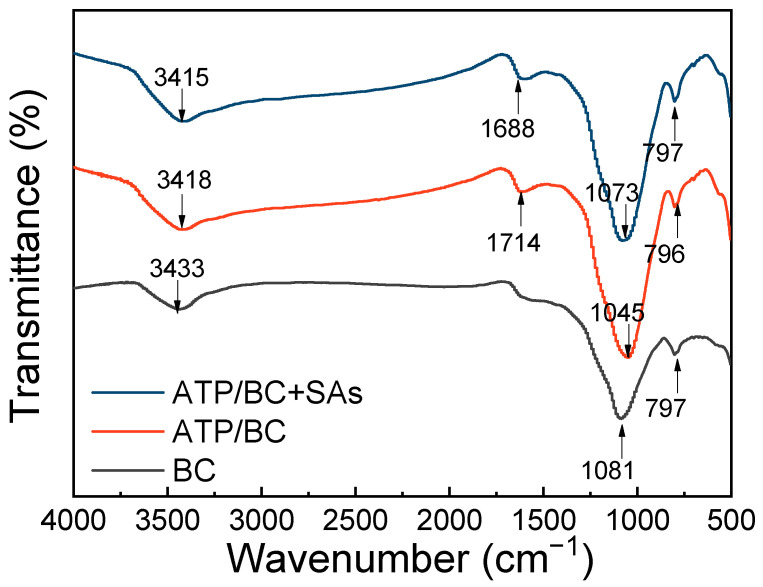
FTIR spectra of ATP, BC, and ATP/BC.

**Figure 4 molecules-27-08076-f004:**
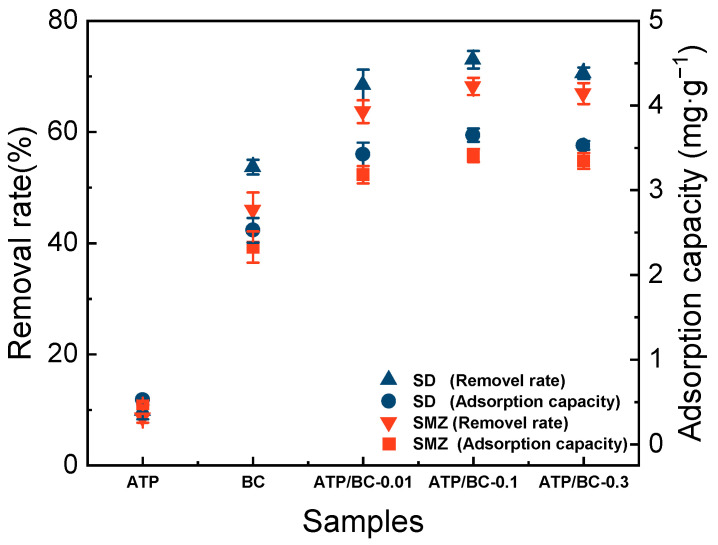
Effect of the mass ratio of ATP to rice straw on the adsorption SD and SMZ.

**Figure 5 molecules-27-08076-f005:**
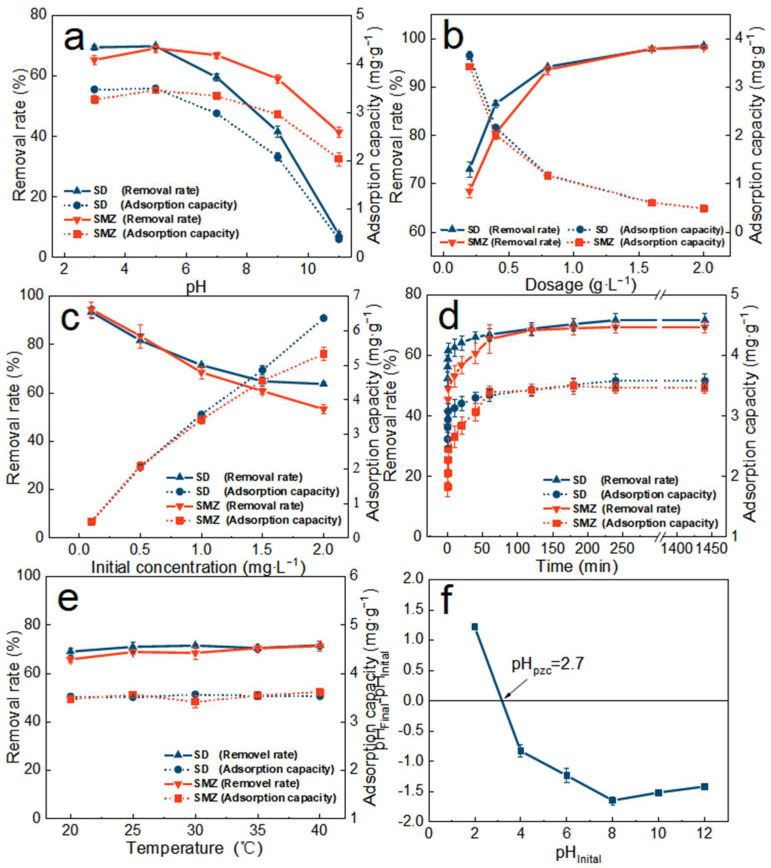
Effect of solution pH (**a**), adsorbent dosage (**b**), initial sulfonamides concentration (**c**), adsorption time (**d**), reaction temperature (**e**) on the adsorption of SD and SMZ by ATP/BC-0.1, and (**f**) measuring of the pH_PZC_ of the ATP/BC-0.1.

**Figure 6 molecules-27-08076-f006:**
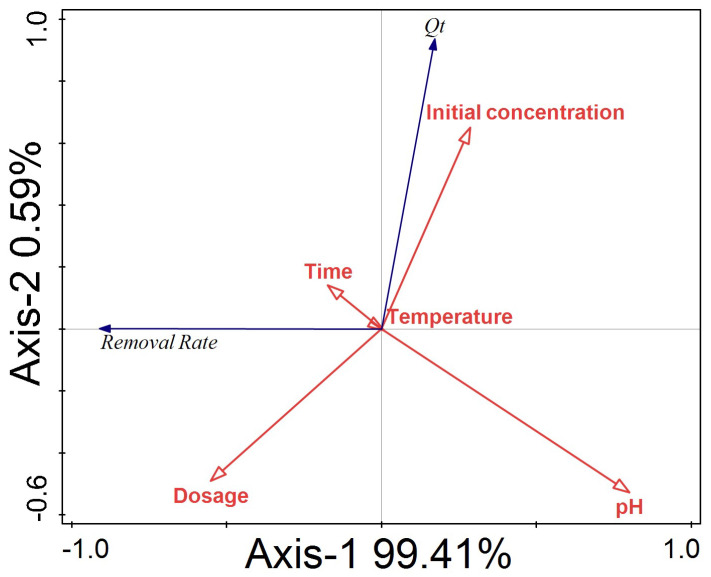
Redundancy analysis of removal rate and adsorption capacity with different environmental factors.

**Figure 7 molecules-27-08076-f007:**
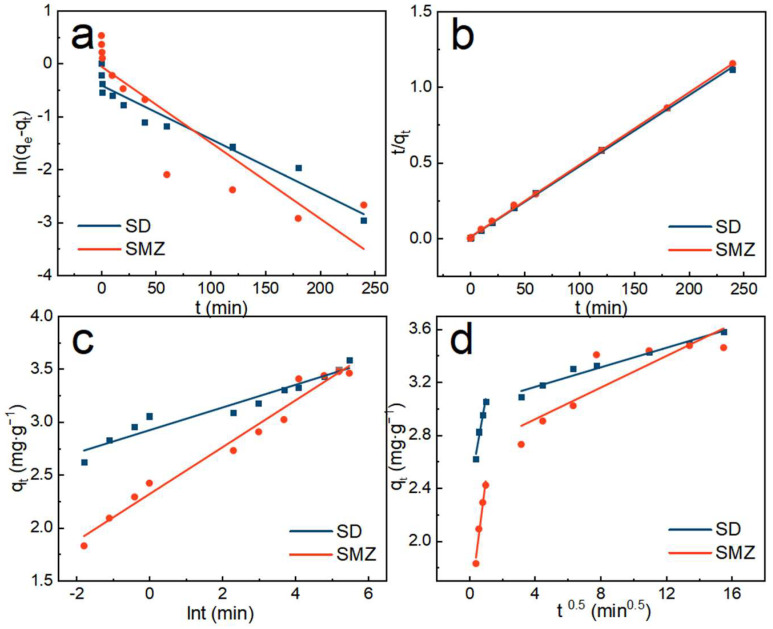
(**a**) Pseudo-first-order, (**b**) pseudo-second-order, (**c**) Elovich, and (**d**) diffusion kinetic models of sulfonamides adsorption by ATP/BC-0.1.

**Figure 8 molecules-27-08076-f008:**
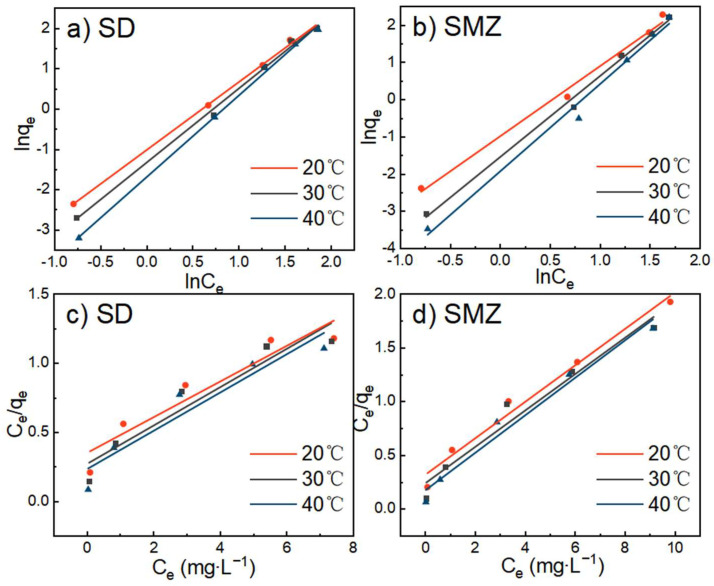
Freundlich (**a**,**b**) and Langmuir (**c**,**d**) isotherms of sulfonamide adsorption by ATP/BC-0.1.

**Figure 9 molecules-27-08076-f009:**
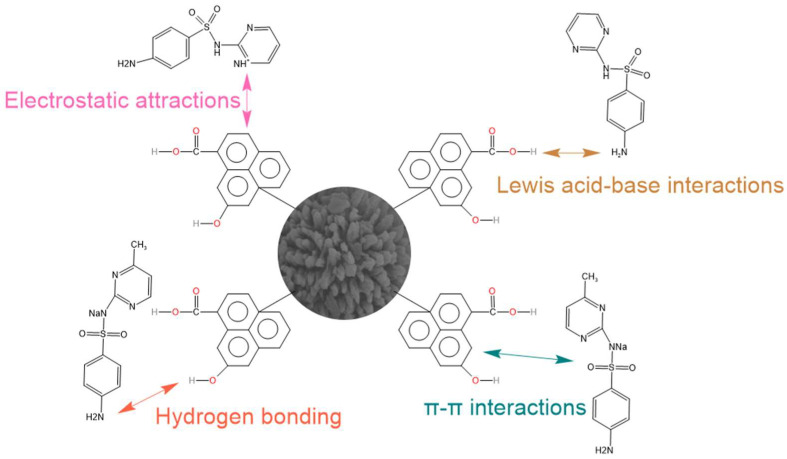
Potential adsorption mechanisms of sulfonamides by ATP/BC-0.1.

**Table 1 molecules-27-08076-t001:** The specific surface area and pore size of different adsorbents.

Sample	Specific Surface Area (m^2^∙g^−1^)	Adsorption Average Pore Size (nm)
BC	49.20	6.61
ATP	71.91	8.27
ATP/BC-0.01	97.33	4.17
ATP/BC-0.1	113.75	3.01
ATP/BC-0.3	97.36	4.26
ATP/BC-0.5	85.36	4.84

**Table 2 molecules-27-08076-t002:** Kinetic parameters for the adsorption of sulfonamide by ATP/BC-0.1.

Category	Model Parameters	SD	SMZ
Pseudo-first-order kinetic	*q_e_* (mg∙g^−1^)	0.66	0.95
	*k*_1_ (min^−1^)	1.01 × 10^−2^	1.43 × 10^−2^
	*R* ^2^	0.93	0.79
Pseudo-second-order kinetic	*q_e_* (mg∙g^−1^)	3.55	3.49
	*k*_2_ (g∙(mg∙min)^−1^)	11.01	9.60
	R^2^	0.99	0.99
Elovich kinetic	*α* (mg∙g^−1^)	1.07 × 10^−1^	2.21 × 10^−1^
	*β* (g∙(mg∙min)^−1^)	2.92	2.32
	R^2^	0.93	0.97
Diffusion	*k_d_*_1_ (mg∙g^−1^∙min^−0.5^)	7.01 × 10^−1^	9.75 × 10^−1^
	*C*_1_ (mg∙g^−1^)	2.37	1.48
	*R* _1_ ^2^	0.94	0.95
	*k_d_*_2_ (mg∙g^−1^∙min^−0.5^)	3.69 × 10^−2^	5.97 × 10^−2^
	*C*_2_ (mg∙g^−1^)	3.02	2.68
	*R* _2_ ^2^	0.96	0.74

**Table 3 molecules-27-08076-t003:** Isotherm parameters for the adsorption of sulfonamide by ATP/BC-0.1.

Sulfonamides	Temperature	Freundlich Isotherm			Langmuir Isotherm		
		*n*	*k_f_*	*R* ^2^	*q_m_* (mg∙g^−1^)	*k* _l_	*R* ^2^
SD	20 °C	1.69	1.83	0.99	7.74	0.37	0.86
	30 °C	1.84	2.07	0.99	7.20	0.51	0.89
	40 °C	2.03	2.31	0.99	7.23	0.59	0.87
SMZ	20 °C	1.90	1.69	0.99	5.89	0.53	0.97
	30 °C	2.19	2.04	0.99	5.91	0.71	0.94
	40 °C	2.37	2.28	0.98	5.73	1.01	0.96

**Table 4 molecules-27-08076-t004:** Thermodynamic parameters of sulfonamide adsorption by ATP/BC-0.1.

Types of Antibiotics	Δ*G* (kJ∙mol^−1^)			Δ*H* (kJ∙mol^−1^)	Δ*S* (J∙(mol∙K)^−1^)
	20 °C	30 °C	40 °C		
SD	−23.73	−25.07	−26.12	8.79	111.61
SMZ	−23.92	−25.04	−26.15	11.39	119.89

## Data Availability

Not applicable.

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
