# Peer review of "Enhanced Adsorption of Sulfonamides by Attapulgite-Doped Biochar Prepared with Calcination"

_molecules, 2022, doi:10.3390/molecules27228076_

Round 1
Reviewer 1 Report
Line 183. «...the surface of BC was smoother and there were some micro-pores». In the image, I see only one pore - on the left side of the image. Moreover, I'm not entirely sure that this is the pore, and not a surface defect. The Authors should provide a more reliable image, where the micropores of the biochar will be visible.
Line 186. «...were successfully compounded together». The image only shows the formation of new structures on the surface of the coal. Such structures, presumably, can be formed due to manipulations, such as exposure to ultrasound. To prove the modification, the Authors should provide images of the distribution of elements obtained by the EXD method. Also from the image does not provide information on whether the modification occurs within the pore spaces or only on the surface. Authors should revise this manuscript section.
«...the addition of ATP decreased the average pore width of ATP/BC composites». It is not entirely clear from the text whether the results for BC and ATP are given for the virgin samples or subjected to the same manipulations as the composite sorbents? Can the Authors guarantee that the reason for the decrease in specific surface area is the modification and not destruction of the pore space due to sonication and drying? Authors should present a control experiment.
Effects of adsorbent dosage. Line 275-283. The reasoning looks very strange, especially with regard to the static exchange capacity. The fact is that the efficiency of adsorption never reaches 100%, since there is an adsorption equilibrium, i.e. adsorption is balanced by desorption. With an increase in the dosage of the sorbent, the percentage of adsorption logically increases, since the number of adsorption centers to the number of antibiotic molecules increases. At the same time, with an infinite number of adsorption centers to a limited number of molecules, adsorption will not reach 100%, since it will be balanced by desorption - adsorption equilibrium. With an increase in the dosage of the sorbent, it will seem that there is a decrease in the static exchange capacity, however, this is not the case, it is simply that the number of adsorption centers relative to the number of antibiotic molecules is too large to be saturated. The Authors should increase the initial concentration of the antibiotic in the solution to ensure this. Authors should also revise the text significantly.
Line 343-345 «The pseudo-second-order model assumes that chemisorption exists in the adsorption process, which involves covalent bonding through sharing or exchanging electrons between the adsorbent and the adsorbate». The pseudo-first and pseudo-second order models assume that the rate of the adsorption process is limited either by diffusion or directly by the reaction at the exchange centers. However, these models do not allow drawing conclusions about the mechanism of adsorption - chemical, physical, or ion exchange; other approaches are used for this. Authors should revise the text.
Line 354. «Sulfonamides are diffused within the particles...» Authors should revise the text. Activated carbons, as well as aluminosilicates, unlike ion-exchange resins, are not permeable to cations, and especially large organic molecules. Adsorption occurs on the surface of such materials and has different mechanisms. For this reason, with an increase in the specific surface area, the adsorption capacity of coals also increases. The Authors themselves attribute the improvement in adsorption properties to a change in the pore size. For this reason, infra-particle diffusion should be considered and described as diffusion within the pore space.
Line 377-379. «The Freundlich isotherm describes multi-molecular layer adsorption, and the maximum adsorption capacity of the adsorbent increases as the concentration of the adsorbent mass increase». The model of Henry, Freindlich, Langmuir, as well as their derivative models of Sips, Liu are based on the fact that with an increase in the initial concentration of the extracted component, the value of the static exchange capacity increases.
Probably, the Authors do not understand the physical meaning of Freundlich model equation, which is a special case of Langmuir model. The Langmuir equation describes the monolayer adsorption process, but not «multi-molecular layer». If Langmuir model described, while filling by monolayer, the isotherm reached to the maximum and become parallel to X-axis, and Freundlich model described only the initial part of this curve. It is known, that the Henry equation described the adsorption in the ultra-low concentration range: ADS=Kg*C, where Kg – Henry constant, showed the curve slope, and C- it is an equilibrium concentration. If Henry equation will change a little bit, we will get Freundlich equation ADS=Kf*C^n, where Kf –it is Freundlich constant, described the same incline, but the curve’s incline, and degree n – described the deviation from the straight line, those from Henry equation. The Freundlich equation described the adsorption from low and medium concentrations, when the monolayer was not filled, and the parameter n described the heterogeneity of adsorption sites. With multilayer adsorption, the isotherm will be S-curve (S-type), which, for example, can be described with BET equation.
Reviewer 2 Report
1. Page 2 Line 54. It is not true, that biochar is characterized by a high specific surface area. Depending on the type of feedstock used for its production and pyrolysis conditions, the specific surface area of biochar can vary from about 10 to about 300 m2 g-1. When compared to chemically activated carbons (up to 3000 m2 g-1) or MOFs (up to 7000 m2 g-1), this value is rather low. Please revise.
2. Please unify the method of units designation i.e. m2 g-1, mg L-1, instead of m2/g, mg/L etc.
3. Fig. 5f. pHzpc should be pHpzc. Moreover, the value obtained using zeta potential measurement is not the same as the pHpzc! These variables differ from the nomenclature and the method of their measurement (despite they can be familiar in some cases). Please revise.
4. Figure 7d. The intraparticle diffusion model should be shown as qt=f(t0.5) and time in min0.5. Also for other models of kinetic adsorption, the time should be provided in minutes – as was also shown by you in the Point 2.5. Please revise the figures. Are the values of constants (k1 and k2) calculated for minutes or hours? Please use the units in the Table 2 to clarify.
5. Figure 7a. The plot of linear form of pseudo-first-order model is ln(qexp-qt)=f(t)! Not t/qt=f(t) as it is for the pseudo-second-order model! Please revise!
6. Figure 7c. Why the time is a negative value?! Please revise. For Elovich model the time on x-axis should be given as ln(t) in minutes! Than this is accurate value.
7. Please unify the y-axis scale in Figure 7 for all images.
8. Please unify the temperature unit. In some cases you show it in Kelvin in other in Celsius degree. Please revise.
9. In how many replicates were the adsorption experiments performed? Please clearly mention this information in the materials and methods section. Also, please add the error bars to the graphs.
10. You have mentioned, that some parameters have changed ‘significantly’. What statistical test was used to confirm this thesis? If you did not compare the results statistically, please rewrite the manuscript, to avoid the term ‘significantly’.
11. I would recommend to compare the performance of your adsorbents with the literature data – adsorption capacities obtained from Langmuir model, adsorbent dose, adsorption temperature, adsorption pH, initial concentration range, to show that your adsorbents are better and novel, than already published.
12. Please revise the manuscript in terms of grammar and typos errors.
13. Please indicate more clearly the novelty of your work and importance of your research.
14. Please rewrite the conclusions in terms of your obtained results. They should be more clearly shown to the readers.
15. It is rather a personal opinion, but when you show the adsorption capacity and removal rate, rather more common method is to show the adsorption capacity on the left y-axis and the removal rate on the right y-axis. However there is no need to change this in the present work. To make the graphs easier to read, you can show the removal rate only when you have a single-point adsorption (the effect of pH, the effect of adsorbent dose), and when you show the adsorption kinetics you do not have to show the removal efficiency as it can be estimated based on the results shown at the beginning. Just as an example how can you visualise your data you can go through the work: https://doi.org/10.1016/j.jece.2022.108567 (Fig S2.). Please treat this work as an example ONLY (no need to cite this work), as it is out of the scope of your research.
Reviewer 3 Report
The paper "Enhanced adsorption of sulfonamides by attapulgite-doped biochar prepared with calcination" deals with interesting and timely subject. The authors study the effects of raw materials ratio on the adsorption capacity of ATP/BC, the adsorption effect of ATP/BC on sulfonamides under the influence of adsorbent dosage, time, solution pH, initial concentration, and temperature, which provides an optional material to treat sulfonamides in wastewater and groundwater. Therefore, I suggest the paper can be published in molecules after major revision. Some specific comments are as follows:
1. What is the innovation of the article?
2. Isotherms conforming to the Freundlich model do not lead to the conclusion that chemisorption occurs, and it is recommended that kinetics and isotherms be discussed separately.
3. A brief explanation of why the technologies that have been reported are inefficient, environmentally unfriendly and impractical is suggested.
4. What is the basis for choosing the pyrolysis temperature? Why did you choose 700°C at section 2.2.
5. The current FTIR data is a little thin to support the conclusion that the modified material has more functional groups. Additional XPS characterization data can be considered at section 3.1.
6. It would be better to add more BET analysis data, such as adsorption-desorption isotherm type, pore size distribution.
7. The figure numbers in rows 209 and 265 are incorrect.
8. FTIR data analysis lacks reference support.
9. To exclude the influence of the specific surface area of the material on the adsorption capacity, the adsorption amount should be normalized by the specific surface area to prove the best adsorption performance of ATP/BC-0.1.
10. A comparison of the adsorption performance of the currently reported adsorbents for SD and SMZ would better highlight the high efficiency of the materials used in this paper.
11. ΔG < -20 kJ/mol, which generally indicates that the adsorption process is chemisorption dominated, but the value of ΔG was not analyzed in the paper.
12. The conclusion that "Freundlich isotherms correspond to both physisorption and exothermic processes" needs more support in the literature, and we do not seem to have seen such a statement so far.
13. The conflict between isotherm model fit data and thermodynamic data is explained at section 3.6, but no final conclusion is given. It is not clear here whether the adsorption process is chemisorption or physisorption dominated, or whether physisorption and chemisorption coexist, and whether the adsorption is exothermic or endothermic at section 3.6.
14. The hydrogen bonding of the adsorption mechanism is not well discussed.
Round 2
Reviewer 1 Report
I have no additional comments.
Author Response
We would like to thank the editor and reviewer for their very useful and constructive remarks, which give us the opportunity to revise and to improve our manuscript.
Reviewer 2 Report
The Authors have improved their work, however there are still some small improvements needed before publication. I recomend the work to be published after some minor repairs. Please find my comments listed below.
1. 'Elovih' should be 'Elovich'
2. 'decrease of' should be 'decrease in'
3. Figure 5d. The time, as was mentioned in the first revision round should be in minutes! Please revise the graph.
4. Figure 5f. The zeta potential and pHpzc are not the same values! I am aware that this is a common mistake, but this must be clarified. I am also missing the methodology how the value was obtained. You can find some helpfull information in the work: https://doi.org/10.1016/j.molliq.2018.04.030 . You can clearly see, that surface charge density=f(pH) allows to define the pHpzc (one of the methods). On the other hand, zeta potential =f(pH) allows to define pHiep (isoelectric point). Please revise.
5. Please make sure, that the values of kinetic model parameters given in Table 2 were calculated based on time in minutes, not in hours!
6. Table 2 - I would recommend to improve the decimal expansion of obtained data up to 3 nubmers.
Reviewer 3 Report
The Authors have improved their work according to the reviewers’ comments.
Author Response

(The authors gave the same response as above.)
